# Characterisation of the horse transcriptome from immunologically active tissues

Joanna Moreton[1,2,3], Sunir Malla[2], A. Aziz Aboobaker[4], Rachael E. Tarlinton[3] and Richard D. Emes[1,3]

[1] Advanced Data Analysis Centre, University of Nottingham, Sutton Bonington Campus, Loughborough, Leicestershire, UK
[2] Deep Seq, School of Life Sciences, University of Nottingham, Medical School, Queen's Medical Centre, Nottingham, UK
[3] School of Veterinary Medicine and Science, University of Nottingham, Sutton Bonington Campus, Loughborough, Leicestershire, UK
[4] Department of Zoology, University of Oxford, Oxford, UK

## ABSTRACT

The immune system of the horse has not been well studied, despite the fact that the horse displays several features such as sensitivity to bacterial lipopolysaccharide that make them in many ways a more suitable model of some human disorders than the current rodent models. The difficulty of working with large animal models has however limited characterisation of gene expression in the horse immune system with current annotations for the equine genome restricted to predictions from other mammals and the few described horse proteins. This paper outlines sequencing of 184 million transcriptome short reads from immunologically active tissues of three horses including the genome reference "Twilight". In a comparison with the Ensembl horse genome annotation, we found 8,763 potentially novel isoforms.

## INTRODUCTION

While no longer the principal means of transport in much of the world, the horse is still an economically important animal in agriculture, sport and gambling associated with horse racing. Individual stallions may be worth several millions of dollars and attract high stud fees creating considerable interest in the genetics of performance traits (*Hill et al., 2010*). In addition, there are several components of the equine immune system that make them in many ways a better model of some human disorders than the most commonly used rodent models. These include, similarly to humans, an exquisite sensitivity to the effects of lipopolysaccharide (LPS) with associated endotoxemia and sepsis (*Bryant et al., 2007*). However, the immune response of the horse has not been well characterised, largely due to the difficulties in working with large animals in experimental settings.

With the difficulty in working with large animals, there is a lack of expressed sequence tag (EST) data, hence the current annotation of the protein coding regions of the horse

Corresponding author
Joanna Moreton,
joanna.moreton@nottingham.ac.uk

genome is largely derived from extrapolation from the genes of other species (*Coleman et al., 2010*). This is beginning to be redressed with several recent papers outlining transcription profiles using digital gene analysis of a variety of horse tissues including muscle, leukocytes, cartilage, brain, reproductive tissue, embryos, sperm and blood (*Capomaccio et al., 2013*; *Coleman et al., 2010*; *Das et al., 2013*; *Iqbal et al., 2014*; *McGivney et al., 2010*; *Park et al., 2012*; *Serteyn et al., 2010*). *Capomaccio et al. (2013)* identified new putative non-coding sequences within intergenic and intronic regions whilst *Das et al. (2013)* suggested additions to the structural annotation of four sperm genes. Two of the other studies (*Coleman et al., 2010*; *Park et al., 2012*) detailed extensions to the annotated gene catalogue in the horse based on transcriptome analysis of quite differing tissue sets, methods and results to those used in this paper. They show that the actual expressed transcription profile only partially overlaps the annotated gene set. A direct comparison of our and these two studies is difficult due to the differing tissues, methodologies and the lack of available locations of the predicted novel genes from these studies.

To extend this description and annotation of horse transcripts, we focus on immunologically active tissues in the horse. To best identify novel transcripts we have sampled multiple tissues and animals including lymphocytes from Twilight, the animal from which the current horse reference genome is derived. Comparison of this animal with lymphocytes, core immunologically active tissues (lymph node and spleen) and other tissues (liver, kidney and jejunum) from two unrelated animals allows a unique catalogue of the immune system transcriptome.

## MATERIALS AND METHODS

### Samples, library preparation and sequencing

The methods are described fully in our previous work (*Brown et al., 2012*) but briefly, five tissue samples; kidney, jejunum, liver, spleen and mesenteric lymph node were collected (as quickly as the post-mortem allowed) from an aged gelding (castrated male horse) euthanised due to osteoarthritis. The tissue samples were collected from an animal euthanised for clinical reasons, by the veterinary surgeon, under the Veterinary Surgeons act of 1966. Full informed consent of the owner was obtained for use of the samples, taken from that animal post-mortem. Lymphocytes isolated by Ficoll Paque (GE healthcare) from a healthy 11 year old welsh mountain pony gelding were kindly provided by Dr. Julia Kydd (School of Veterinary Medicine and Science, University of Nottingham) under the Home Office and local Ethical Approval Committee (PPL 40/3354). RNA extraction on these samples was performed using the Nucleospin RNA II mini kit (Machery Nagel) according to manufacturer's instructions.

RNA from lymphocytes isolated from a healthy adult Thoroughbred mare ("Twilight" *Wade et al., 2009*) was kindly provided by Donald Miller (Baker Institute of Animal Health, Cornell University, USA). Total RNA was isolated from snap frozen lymphocytes using the RNeasy kit (Qiagen, Valencia, CA, USA) following the manufacturer's protocol. This horse (born in 2004) was maintained at the Baker Institute for Animal Health, Cornell University, Ithaca, NY, USA. Animal care and research activities were performed

**Table 1  Read statistics for the seven samples.**

| Sample | Horse | Raw reads | Trimmed reads | % of raw trimmed | Reads aligned | % of trimmed aligned | Average coverage[*] |
|---|---|---|---|---|---|---|---|
| Lymphocyte A | A | 20,853,992 | 18,243,283 | 87% | 7,856,017 | 43% | 0.15 |
| Lymphocyte B | B | 32,050,093 | 27,315,182 | 85% | 11,659,787 | 43% | 0.23 |
| Jejunum | C | 19,902,170 | 17,241,772 | 87% | 7,659,938 | 44% | 0.15 |
| Kidney | C | 33,158,285 | 27,746,321 | 84% | 10,937,750 | 39% | 0.21 |
| Liver | C | 23,176,545 | 19,982,256 | 86% | 8,565,159 | 43% | 0.17 |
| Lymph node | C | 24,671,029 | 21,444,476 | 87% | 9,221,340 | 43% | 0.18 |
| Spleen | C | 30,421,675 | 26,828,834 | 88% | 12,708,499 | 47% | 0.25 |

Notes.

A, "Twilight", healthy Thoroughbred; B, healthy castrated male welsh mountain pony; C, aged gelding euthanised for arthritis.

[*] Based on the number of base pairs in Ensembl v71 genome assembly (2,428,790,173bp) and average read length after trimming (47bp). Shown to two decimal places.

in accordance with the guidelines set forth by the Institutional Animal Care and Use Committee of Cornell University, protocol # 1986–0216, approved until March 2013. Although a member of the research herd at the Equine Genetics Center, Twilight was never a participant in any of the experimental activities. Her main contribution to research is through blood samples for experiments using DNA and RNA.

The RNA derived from the tissue samples was used as the starting material for sequencing. This was performed on a SOLiD 3 ABI sequencer generating 50 bp reads according to the manufacturer's instructions. Read data are available at the EBI Sequence Read Archive (SRA) under the study accession number ERP001116.

## Read trimming and alignment

The horse genome assembly EquCab2 (*Wade et al., 2009*) was downloaded from Ensembl v71 (www.ensembl.org) and contained 26,991 genes and 29,196 transcripts. CLC Genomics Workbench version 6 (CLC Bio, Aarhus, Denmark, www.clcbio.com) was used to apply quality, SOLiD adapter and Poly-N trimming to the read sequences (File S1). The limit for the removal of low quality sequences was set at 0.2 and a maximum of two ambiguous nucleotides were permitted in each sequence. In CLC each quality score is converted to an error probability where low values represent high quality bases. For each base the error probability is subtracted from the limit (0.2 here). The cumulative total of this value (limit—error) is calculated for each base and it is set at zero if it becomes negative. The retained part of the read will start at the first positive value and end at the highest value of the cumulative total. Any reads less than 20 bp were removed after trimming and the average read lengths were 47 bp. The average coverage values (number of reads *x* read length/genome size) for each sample based on the aligned reads are shown in Table 1.

TopHat 2.0.9 (*Trapnell, Pachter & Salzberg, 2009*) was used to align the reads to the repeat masked version of the horse genome (Ensembl v71) to enable non-redundant transcriptome analysis. TopHat first aligns non-spliced reads using Bowtie 1.0.0 (*Langmead et al., 2009*) then identifies splice junctions. Gapped alignments are then used

**Peer**J

by TopHat to map the reads not aligned by Bowtie. In order to utilise the splice sites in all samples, two iterations of TopHat alignments were carried out (*Cabili et al., 2011*). Firstly, the reads from each sample were aligned to the repeat-masked horse genome with default parameters and the option to incorporate genome annotation (parameter "–GTF") was not used. The splice sites ("junctions") were extracted from all of the output files and duplicates were removed leaving 216,007 sites. These splice sites were pooled together with the non-redundant sites extracted from the Ensembl annotation yielding 399,264 non-redundant splice sites. Each of the samples were then re-aligned with TopHat using the pooled non-redundant splice sites file (with 'raw-juncs' and 'no-novel-juncs' parameters) to the repeat-masked genome. By using the splice sites from the first iteration of TopHat and also Ensembl we generate a transcriptome using a combination of *de novo* and annotated information.

TopHat was used for the read alignment because it is part of the Tuxedo suite and is therefore a natural input for the Cufflinks assembler (*Trapnell et al., 2010*). It is also the preferred aligner for Scripture (*Guttman et al., 2010*). Cufflinks and Scripture are described in the transcriptome assembly section.

## Transcriptome assembly

Each of the samples were assembled into separate transcriptomes using two different "mapping first" tools; Cufflinks v2.1.1 (*Trapnell et al., 2010*) and Scripture (*Guttman et al., 2010*) (beta2 version, December 2010). These tools both require the reads to be aligned to a reference genome first but use different approaches for transcript assembly. A minimal set of transcripts is assembled by Cufflinks using a probabilistic model. It performs a minimum cost maximum matching in bipartite graphs (*Trapnell et al., 2010*). Scripture however creates a connectivity graph which represents the adjacency that occurs in the RNA but that is broken in the genome by an intron sequence. A statistical segmentation strategy is used to determine paths with aligned read enrichment over background noise (*Guttman et al., 2010*).

Both Cufflinks and Scripture were run using default parameters, however due to computational time Scripture was run on the named chromosomes only (not on the unanchored contigs "chrUn"). The samples were assembled individually to reduce the complexity of isoforms and hence reduce the chance of incorrectly assembled transcripts (*Trapnell et al., 2012*). The Cufflinks and Scripture assembly files are provided as File S2 and File S3.

The "Cuffmerge" program (included in the Cufflinks package) was used to merge the Cufflinks and Scripture assemblies separately. Stranded transcripts from the two assemblies were compared using the Cufflinks inclusive program "Cuffcompare" with the Cufflinks assembly as a mock reference. The class codes in the Cuffcompare output were used to generate a consensus assembly (University of Nottingham "UoN", File S4). This consensus assembly was compared to the Ensembl annotations using Cuffcompare (File S5).

## Annotation

The UoN cDNA sequences (File S6) were extracted from the consensus assembly (*gtf) file and the longest open reading frames (ORFs) were determined. Gene annotation was conducted by prediction of Pfam domains (PfamA.hmm library downloaded June 2013) (*Punta et al., 2012*) using HMMER (*Eddy, 2011*). Associated gene ontology (GO) terms (*Ashburner et al., 2000*) were determined using the Pfam2GO database (version compiled 15/6/2013) of InterPro (*Hunter et al., 2009*). The UoN transcripts were searched against the NCBI non-redundant (NR) database (downloaded 14th November 2013) using BLASTX (*Altschul et al., 1997*), a cutoff evalue of 1e-10 was used to infer homology.

## Gene expression analyses

The TopHat BAM files were filtered for unique alignments (SAM flag NH:i:1) and the number of tags per Ensembl gene was calculated using htseq-count (*Anders, Pyl & Huber, 2014*). These counts were converted into Reads per Kb per million (RPKM) values (*Mortazavi et al., 2008*). A table of RPKM values for all Ensembl genes is provided as File S7.

   As the number of replicates was limiting, identification of genes differentially expressed between samples was not attempted. However, genes enriched in each sample were identified as those expressed above a simple threshold. The threshold was determined using the following criteria; RPKM > 5 within a sample (to ensure robust expression within the test sample) and an RPKM above the threshold (RPKM >10 × the mean of RPKMs for the other samples) (File S8). The samples are described in Table 1. The "hclust" command in R (*R-Core-Team, 2013*) was used for the hierarchical clustering analysis of gene expression values (RPKMs). It was performed using the default complete linkage method and Euclidean distance. Probability values for each cluster were calculated using the "pvclust" R package (*Suzuki & Shimodaira, 2006*) (bootstrap $n = 1000$).

## Comparison of horse and human gene families

To identify orthologous and potential paralogous gene expansions in the horse evident in our transcriptome data, translations of the longest ORF of all predicted horse transcripts were compared to proteins encoded by known human genes (Ensembl build GRCh37.71). Both human and horse proteome sets were first clustered to collapse within-species identical protein sequences generated from alternative transcripts using CD-HIT (*Li & Godzik, 2006*). This resulted in 64,231 human and 29,090 horse sequences. These were compared using Inparanoid (version 4.1, overlap cutoff = 0.5, group merging cutoff = 0.5, scoring matrix BLOSUM62) (*Remm, Storm & Sonnhammer, 2001*). Functional comparison of gene sets was conducted using Ingenuity Pathway Analysis (Ingenuity Systems).

# RESULTS

## Transcriptome assemblies

Around 184 million reads were generated and 159 million remained after trimming; approximately 68.6 million of which were aligned to the reference genome EquCab2

(Table 1). Scripture assembled 102,270 stranded transcripts (27,610 with >1 exon, File S3) whereas Cufflinks reconstructed 58,182 (20,459 with >1 exon, File S2). There were 10,518 Cufflinks transcripts that completely matched the intron chain of the Scripture transcripts. In addition to this 18,152 Cufflinks transcripts contained or covered at least one Scripture transcript with the same compatible intron structure (File S9; Venn diagrams generated with R package "venneuler" *Wilkinson, 2011*). The union of these two sets resulted in 28,230 transcripts, 14,762 of which contained more than one exon (File S4).

## Comparison of consensus assembly to Ensembl

The similarities between the 28,230 consensus transcripts (henceforth referred to as "UoN", University Of Nottingham) and the 28,944 Ensembl transcripts on the named chromosomes were compared (File S5). There were only 507 UoN transcripts which completely matched the intron chain of an Ensembl transcript (File S9). The majority of transcripts (8763, 31%) were identified as potentially novel isoforms of a predicted Ensembl transcript with at least one splice junction shared.

The majority of Ensembl transcripts (18668, 65%) did not overlap with a UoN transcript (File S10). This could be due to the strict consensus approach used for the UoN assembly. Also, the specific tissues analysed would not be expected to reconstruct all the transcripts from Ensembl, which are predicted from genomic DNA, and hence all potential transcriptomes not those limited to the tissues we have analysed here.

Around 9,500 (34%) of the 28,230 UoN transcripts were annotated with a Pfam protein domain, approximately 6,600 (23%) with at least one GO term and 16,166 (57%) had at least one significant BLASTX hit against NCBI-NR (File S11). In total there were 16,305 UoN transcripts with at least one annotation. The UoN annotated transcripts were split into Cuffcompare categories based on the comparison to the Ensembl annotations (File S11). As expected, the transcripts matching the intron chain ("=") or sharing at least one splice junction ("j") of the Ensembl annotations had the highest percentage of annotated transcripts (e.g., 97% and 99% with BLASTX hits respectively). There were 367 of the 16,166 UoN transcripts with a BLASTX hit that showed homology to only a single species and just under half of these (163) were to *Equus caballus*. The top hit was extracted for each transcript and as expected most of these hits were also to the *Equus caballus* genome. Other mammals with a high number of top hits were *Homo sapiens, Mus musculus, Ceratotherium simum simum, Tursiops truncatus* and *Sus scrofa*. The full list is shown in File S12.

## Gene expression analyses

The number of Ensembl genes specific to each sample is shown in Table 2 and File S8 (see also materials and methods). By our strict criteria, no genes were enriched in more than one sample. The Lymphocyte A sample had many more specific genes than Lymphocyte B. This is possibly due to sample A being taken from the same horse that the published genome is derived from, however the read alignment rate between these two samples is similar suggesting this may not be the major factor. Alternatively this may reflect the immune states of individual horses at the time of sample collection.

**Table 2 Number of sample-enriched genes.**

| Sample | # Sample-enriched genes |
| --- | --- |
| Lymphocyte A | 201 |
| Lymphocyte B | 23 |
| Jejunum | 228 |
| Kidney | 318 |
| Liver | 272 |
| Lymph node | 44 |
| Spleen | 79 |

**Notes.**
(A) "Twilight", healthy Thoroughbred (B) healthy castrated male welsh mountain pony.

The top ten gene ontology (GO) terms for the sample-enriched genes largely reflect the known function of the tissues sampled (File S13). Hierarchical clustering analysis of the RPKMs between tissues showed three clades (Fig. 1). The branch values are the pvclust approximately unbiased (AU) $p$-values (left) and bootstrap (BP) probability values (right) where the $p$-values are expressed as percentages (95% is equivalent to $p$-value $< 0.05$) (*Beliakova-Bethell et al., 2013*). For each of the nodes, the AU bootstraps are over 80% and these are reported having superiority over the BP values (*Suzuki & Shimodaira, 2006*). The lymphocyte samples cluster most closely with the spleen sample which likely reflects the high number of lymphocytes present in the spleen at the time of collection. Whilst the kidney and liver have general shared roles in waste excretion suggesting a possible overlap of transcription profile, determining a definitive reason for the separation of the clade containing lymph node, kidney and liver is not clear. The jejunum sample forms an outgroup and this separation from the other immune-like tissues likely reflects the relatively smaller proportion of lymphoid (Peyer's patch) tissue to non-lymphoid material in this organ. It is also important to consider that only a limited number of samples and animals are compared and so robustness of these relationships is not ensured.

Analysis of genes enriched in each sample identified enriched canonical pathways. The kidney sample is enriched in genes involved in the "$\gamma$-glutamyl Cycle", "Leukotriene Biosynthesis", "Glycine Cleavage Complex", "$\beta$-alanine Degradation I" and "4-hydroxyproline Degradation I" pathways. Amino-acid catabolism pathways, possibly reflecting high-energy consumption of the kidney, dominate these. The liver sample is enriched with genes involved in the degradation of chemical products (e.g., nicotine and melatonin). Enzymes including members of the CYP450 and UDP-Glucuronosyltransferase (UGT) gene families, which are known to be highly expressed in the liver, are enriched. The spleen shows enrichment of genes involved in the pathways "Autoimmune Thyroid Disease Signaling", "Hematopoiesis from Pluripotent Stem Cells", "Primary Immunodeficiency Signaling", "Dendritic Cell Maturation", and "Agranulocyte Adhesion and Diapedesis". Largely these are due to the enrichment of genes encoding the immunoglobulin heavy chain and Fc fragment of IgG. Enrichment of these pathways reflects the role of the spleen as a primary site of white blood cell differentiation and

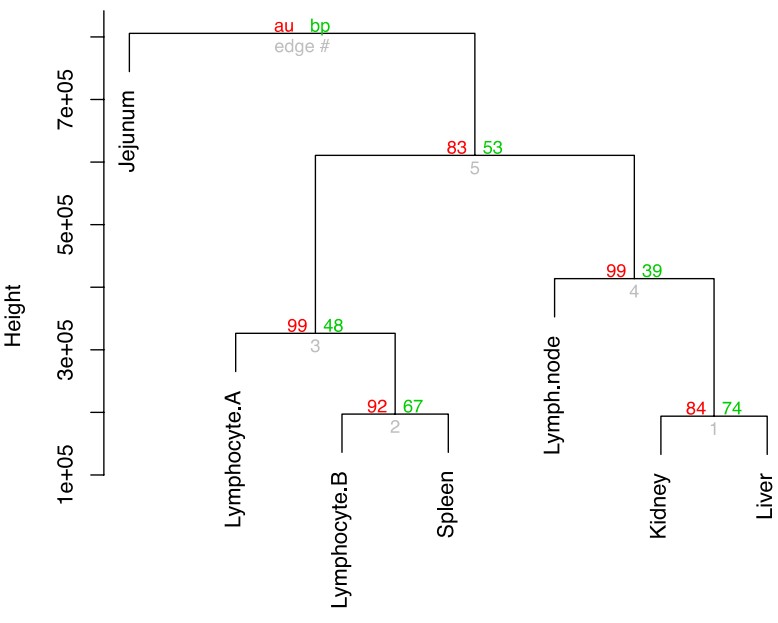

**Figure 1 Hierarchical clustering of gene expression profiles in seven tissues.** The R command "hclust" was used for the hierarchical clustering analysis. The branch values are the pvclust approximately unbiased (AU) *p*-values (left) and bootstrap (BP) probability values (right) where the *p*-values are expressed as percentages.

storage. The lymph node sample is enriched in the pathways, "Primary Immunodeficiency Signaling", "Hematopoiesis from Pluripotent Stem Cells", "Autoimmune Thyroid Disease Signaling", "Allograft Rejection Signaling" and "Communication between Innate and Adaptive Immune Cells". As with the spleen these are predominantly due to the enrichment of genes encoding the immunoglobulin heavy chain proteins and result from the white blood cell content contained in the tissue.

## Identification of paralogous gene expansions in horse

Previously the horse genome was described as containing lineage specific expansions of olfactory and immune genes (*Wade et al., 2009*). The expansion of these families particularly immune related genes is often seen in mammalian genome comparisons (*Emes et al., 2003*). *Wade et al. (2009)* reported that there were 99 gene families expanded in the horse genome. Comparison of the proteins encoded by the transcripts found here identified 4,605 groups of horse:human orthologs and 10,607 in-paralogs. The majority of these represent expansions in human where a single horse protein was encoded by the transcriptome data generated here. 91 families were identified with a specific expansion

in horses (many:1 relationship). Of these the large majority (83/91) represent simple duplications in the horse transcriptome compared to human. Three families have four non-identical encoded proteins orthologous to a single protein in humans. Annotation of these genes identifies them as T cell receptor alpha constant (TRAC), heparin sulfate proteoglycan 2 (HSPG2) and solute carrier family 23 (ascorbic acid transporter) member 1 (SLC23A1). An additional four gene families are identified with three encoded proteins in horse compared to a single protein in human. These are GTPase, IMAP family member 7 (GIMAP7), UDP glucuronosyltransferase 1 family polypeptide A6 (UGT1A6), solute carrier family 44 (SLC44A2), ATP-binding cassette, sub-family C member 8 (ABCC8 and sushi, nidogen and EGF-like domains 1 (SNED1).

An additional 99 families were found with expansions in both human and horse (many:many relationship). Reflecting the tissues used for RNA extraction, genes in this category are highly enriched for immune functions. The most significantly populated pathways are "Role of NFAT in regulation of the immune response", "CD28 Signaling in T helper cells", "iCOS-iCOSL signaling in T helper cells", "Natural killer cell signaling" and "PKC$\theta$ signaling in T lymphocytes".

## DISCUSSION

The analysis conducted here provides insight into the transcriptome of immune tissues from the horse and makes these analyses freely available (Supplemental Files). Whilst it is unclear why the horse transcriptome should contain the specific expansions of gene families described, the analysis provided insight into potential areas of T-cell biology which may underlie equine specific immunobiology. The analysis conducted also allowed the identification of gene expansions such as UGT1A6, part of a putative paralogous gene expansion in horse relative to human. UGT1A6 is a member of the UDP-glucuronosyltransferases (UGTs), a gene family essential for metabolism of both xenobiotic and endobiotic substances. In contrast to humans and model organisms, there is currently little information regarding specific drug metabolism in animals of veterinary importance. This is particularly true in the horse. Due to the broad application of its mechanisms on xenobiotic substances, the UGT enzyme group has important implications in pharmacokinetics, the development of drugs and their associated elimination rates. Importantly, as many of the drugs used in equids are adopted from those designed from human UGT research, understanding the differences in genes encoding these proteins may provide a basis for investigation into the UGT group of enzymes in horses and will open up further opportunities for specific pharmacokinetic research into UGT related equine drug metabolism potentially reducing toxic drug interactions.

The data presented here demonstrated the utility of second generation sequencing in significantly advancing knowledge of gene transcription in a poorly characterised species. A large number of potential novel genes were identified alongside some extensions to existing genes. The completeness of these predictions remains to be confirmed by traditional mRNA isolation and sequencing but the data presented provides a valuable resource, freely available for study of equine biology.

## ACKNOWLEDGEMENTS

We thank Dr. Julia Kydd (School of Veterinary Medicine and Science, University of Nottingham) and Dr. Donald Miller (Baker Institute of Animal Health, Cornell University, USA) for their kind donation of lymphocyte samples and RNA. We would also like to thank Dr. Martin Blythe, Damian Kao, Victoria Wright and Katharine Rangeley (University of Nottingham) for useful discussions.

### Funding

Funding for this project was provided by the University of Nottingham. The funders had no role in study design, data collection and analysis, decision to publish, or preparation of the manuscript.

### Grant Disclosures

The following grant information was disclosed by the authors:
University of Nottingham.

### Competing Interests

Richard D. Emes is an Academic Editor for PeerJ.

### Author Contributions

- Joanna Moreton conceived and designed the experiments, analyzed the data, wrote the paper, prepared figures and/or tables, reviewed drafts of the paper.
- Sunir Malla performed the experiments, wrote the paper, reviewed drafts of the paper.
- A. Aziz Aboobaker conceived and designed the experiments, wrote the paper, reviewed drafts of the paper.
- Rachael E. Tarlinton contributed reagents/materials/analysis tools, wrote the paper, reviewed drafts of the paper.
- Richard D. Emes conceived and designed the experiments, analyzed the data, wrote the paper, reviewed drafts of the paper.

### Animal Ethics

The following information was supplied relating to ethical approvals (i.e., approving body and any reference numbers):

(1) Home Office and local Ethical Approval Committee (PPL 40/3354) (2) Institutional Animal Care and Use Committee of Cornell University, protocol # 1986-0216.

### DNA Deposition

The following information was supplied regarding the deposition of DNA sequences:

Read data are available at the EBI Sequence Read Archive (SRA) under the study accession number ERP001116.

## Supplemental Information

Supplemental information for this article can be found online at http://dx.doi.org/10.7717/peerj.382.

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
