# Peer review of "Characterisation of the horse transcriptome from immunologically active tissues"

_PeerJ, doi:10.7717/peerj.382_

## Round 0.1 · original submission · Major Revisions

After discussion with the publisher, we decided to change the decision of your manuscript from "Reject" to "Revision". Please revise the manuscript accordingly and submit it again. Thanks for your understanding.

Reviewer 1 ·

Basic reporting

The English used in the paper is very poor making it difficult to understand in places. For example, page 3, line 57, the expression “DNA the horse genome is derived from “is confusing.

Experimental design

The study design is flawed in that the starting materials for sequencing were obtained from different sample types of 3 different horses, two were old aged (11 years) (one clearly with osteoarthritis and one healthy). This design includes so many compounding factors on the subsequent transcriptome data: age, disease state, and tissue difference. This biased design made the subsequent transcriptome analysis invalid. Second, the authors adopted an inappropriate method to anazlye the data, transcriptome reconstruction with Scripture and Cufflinks require align all the reads to a reference (unannotated) genome, but the Ensembl v71 is an annotated one.

Validity of the findings

As stated in the Experimental Design section, the biased study design and inappropriate methodology made the whole paper invalid.

Additional comments

The authors investigated the transcripome reads from immunologically active tissues, lymphocytes, and RNA sample of 3 different horses. The researchers observed 8.763 potentially novel isoforms by comparing with Ensembl horse genome annotation. Despite the interesting results, it also had a number of limitations.

1. Page 3, line 48. Please specify the horse age and justify why aged horse was used. The relatively old age implies that some results might be related to degenerative processes or intrinsic response to the pathogenesis of osteoarthritis.
2. Page 3, line 52. Please specify the time interval between euthanize and harvesting tissue, since this time period is critical for tissue preparation, especially for lymphatic tissues.
3. Page 3, line 53, please provide the information how the RNA was prepared from these lymphocytes.
4. Page 3, lines 56-59, please specify the horse age and any potential research activities done would potentially affect the overall and lymphatic system. Apparently, this horse was used in research setting.
5. Page 3, line 62, what were the starting materials for sequencing, lymphocytes, tissue, or RNA? Since the above three different samples were described in the preceding section.
6. Page 3, lines 73-74, “align the reads to … (Ensembl v71)” is not an appropriate strategy to do transcriptome reconstruction with Scripture and Cufflinks. Since these two strategies align all the reads to a reference (unannotated) genome, but the Ensembl v71 is annotated one. See http://www.ncbi.nlm.nih.gov/pubmed/21572440
7. Page 3, line 89, mapping-first approach are the appropriate name for Scripture and Cufflinks.
8. Page 4, line 126, please justify why different RPKM were used for different samples. In addition, the authors could provide the horse and sample information for each RPKM clearly here, this will help potential readers understand the project, since no such information was found in the supplemental file 8.
9. Page 4, line 133, “… translations of all horse transcripts were …” is confused, please specify how you got the translations, by prediction based on transcriptome?
10. Here is a minor comment. The English used in the paper is also very poor making it difficult to understand in places. For example, page 3, line 57, the expression “DNA the horse genome is derived from “is confusing.

Reviewer 2 ·

Basic reporting

The article is well written and procedures sound.
The experimental design seem to be appropriate for the Authors' aim.
My only concern about this paper is about the importance of the findings.
I do not see a major improvement in the field.

Experimental design

Nothing to say.

Validity of the findings

Findings in my opinion are the weak part of this work, but I understand that a new annotation could be of help for the horse community that lacks a robust expression sequence tags repository.

Additional comments

Why the Authors masked the genome for the alignment process?
Why the Authors did not use data from other papers that used the same technology in immune related tissues? They could improve their annotation using more data.

Reviewer 3 ·

Basic reporting

No Comments

Experimental design

No Comments

Validity of the findings

No Comments

Additional comments

The authors analysis the horse transcriptome from immunologically active tissues. As horse is a large animal, few previous studies have been carried out, especailly in the immunologically active tissues. Therefore this study presented new transcriptome data to allow a unique insight into the expression of genes with a functional role in the immune system. The drawback of the paper is that there is no replicates and the identified genes needs to be further validated; further studies should be carried out to validate the genes, even a few of them.

I have the following concerns:
(1) In Table 1, please add (%) to "Trimmed reads" and "Reads aligned" columns. More importantly, please report average coverage for each sample based on the aligned BAM file.
(2) Line 69: "The limit for the removal of low quality sequences was set at 0.2". The authors should make it clear about the meaning of threshold.
(3) Line 142 to line 157: Please make a main figure as a venn diagram, which will clearly show the relationship between the Cufflinks and Scripture derived results, as well as the "UoN" vs Ensembl transcripts.
(4) Line 223: "inparalogs" typo?
(5) Line 229: "(HSPG2" missing parenthesis
(6) Line 239: unknown character after "PKC"

---

## Round 0.2 · accepted · Accept

Dear authors,

After reviewing your revisions, we decide to accept your manuscript for publication.